# Maternal Protein Restriction in Rats Alters Postnatal Growth and Brain Lipid Sensing in Female Offspring

**DOI:** 10.3390/nu15020463

**Published:** 2023-01-16

**Authors:** Valentine S. Moullé, Morgane Frapin, Valérie Amarger, Patricia Parnet

**Affiliations:** Nantes Université, INRAE, UMR 1280, PhAN, IMAD, F-44000 Nantes, France

**Keywords:** maternal protein restriction, perinatal, leptin, adiponectin, food behaviour, lipid sensing, rat

## Abstract

Perinatal nutrition is a key player in the susceptibility to developing metabolic diseases in adulthood, leading to the concept of “metabolic programming”. The aim of this study was to assess the impact of maternal protein restriction during gestation and lactation on glucose homeostasis and eating behaviour in female offspring. Pregnant rats were fed a normal or protein-restricted (PR) diet and followed throughout gestation and lactation. Body weight, glucose homeostasis, and eating behaviour were evaluated in offspring, especially in females. Body weight gain was lower in PR dams during lactation only, despite different food and water intakes throughout gestation and lactation. Plasma concentration of leptin, adiponectin and triglycerides increased drastically before delivery in PR dams in relation to fat deposits. Although all pups had identical birth body weight, PR offspring body weight differed from control offspring around postnatal day 10 and remained lower until adulthood. Offspring glucose homeostasis was mildly impacted by maternal PR, although insulin secretion was reduced for PR rats at adulthood. Food intake, satiety response, and cerebral activation were examined after a lipid preload and demonstrated some differences between the two groups of rats. Maternal PR during gestation and lactation does induce extrauterine growth restriction, accompanied by alterations in maternal plasma leptin and adiponectin levels, which may be involved in programming the alterations in eating behaviour observed in females at adulthood.

## 1. Introduction

In 2015, more than 20 million babies were born with a low birth weight (LBW), i.e., less than 2500 g, around 1 of 7 of all births worldwide [1]. Epidemiological and experimental studies on various animal models have shown that a LBW as a result of intrauterine growth restriction (IUGR) increases the risk of developing metabolic disorders, such as coronary heart disease, hypertension, or type 2 diabetes in adulthood, leading to the concept of developmental origins of health and disease (DOHaD) [2,3,4,5]. However, mechanisms responsible for this metabolic programming remain to be further investigated because very little is known about disturbances in hormone regulation and nutrient utilization in LBW infants that could affect their body composition and eating behaviour. Yet, early changes in leptin and adiponectin levels, which are both secreted by adipose tissue, have been shown in LBW infants and are related to maternal pre-gestational body mass index, birth weight, and offspring postnatal weight gain [6,7,8,9]. IUGR, a possible cause of LBW, is associated with persistent changes in the preference for palatable foods, an altered hedonic response to sweet stimuli in childhood [10,11], a preference for carbohydrates for young IUGR women [12], and LBW, which itself is associated with increased fat intake in school-aged boys [13]. Altogether, these particularities of eating behaviour may lead to the development of metabolic diseases in adulthood.

IUGR animal models have been useful to describe some dysregulation of the homeostatic control of energy balance. The energy homeostasis is finely regulated by the hypothalamus, which is the major centre of convergence and integration of multiple nutrient-related signals [14]. Maternal undernutrition (50% caloric intake) or malnutrition (low protein maternal diet) drastically reduce the postnatal surge of plasma leptin in offspring that has been shown to be a key step necessary for an efficient wiring between the different hypothalamic nuclei of the melanocortin system and a precise regulation of hypothalamic neuropeptides gene expression involved in food intake control and lipid sensing [15,16,17,18,19,20]. In the case of perinatal protein restriction (PR), a comparative genome-wide transcriptomic analysis performed on male rats revealed that the expression of several genes involved in insulin and leptin signaling and in the detection and use of lipid nutrients differed from control animals until adulthood [21], and it was associated with modifications of meal pattern in PR rats [22] and preference for palatable foods [23]. However, the impact of maternal PR on food behaviour remains poorly documented in female offspring.

The central nervous system plays a key role in the regulation of the energy balance in mammals because signals from the periphery, such as hormones, gastrointestinal peptides, and nutrients (glucose, fatty acids), are detected by specialized neurons of the hypothalamus and the brainstem [14,24]. Fatty acid sensing by specific neurons in the lateral hypothalamus [25] and other brain regions, such as mesolimbic systems, seems to be an important component of energy balance regulation. Indeed, (i) short-term lipid infusion toward the brain reduces food intake in rat [26]; (ii) intracerebroventricular administration of oleic acid markedly inhibits food intake and decreases the hypothalamic expression of neuropeptide Y, an orexigenic peptide, in rat [27]; and (iii) the satietogenic effects of lipids involve a direct modulation of the hypothalamic neuronal activity on distinct neuronal populations [28]. Because the architectural organization of the hypothalamus depends on transcription factors and trophic signals, such as insulin and leptin, which are known to be altered in maternal PR [17,20,29,30], its ability to sense lipids and control food intake may be impaired and needs to be examined.

Thus, we hypothesized that altered eating behaviour previously demonstrated in offspring from PR mothers can be related to their brain capacity to sense lipids and respond to the satietogenic effect of lipids. To this end, we used a rat model of maternal PR during gestation and lactation to generate PR and control (CTL) offspring, on which we tested the inhibitory effect of a lipid preload, after a short fast, on their food intake. In addition, hormonal status during the gestation, growth, and glucose regulation were compared between groups in order to decipher early determinants of the programming of eating behaviour.

## 2. Materials and Methods

*Ethics statement.* All experiments were carried out in accordance with the guidelines of the National Legislation on Animal Care of the French Ministry of Research (Decree No 2013-118, 1 February 2013) and with EU legal frameworks relating to the protection of animals used for scientific purposes (i.e., Directive 2010/63/EU). They were approved by the Animal Ethics Committee of Pays de La Loire under reference APAFIS#13253-2020060814013364 v1. The scientists in charge of the experiments received training and personal authorization. The experiment was conducted in the experimental facilities at Nantes University (permit number E-44-015), delivered by French veterinary services the 18 April 2019.

*Experimental diets.* Diets were manufactured at INRAE SAAJ (Jouy-en-Josas, France), and their compositions are provided in Table 1. Control diet contained 20% protein, while low-protein diet contained 8% protein to induce a maternal PR during gestation and lactation.

*Animals*. Pregnant Sprague–Dawley rats were purchased at gestation day 1 (G1) from Janvier Labs (Le Genest Saint Isle, France). On arrival, rats were individually housed under controlled conditions (22 °C, 12 h/12 h dark/light cycle, red tunnel and Kraft paper curl for enrichment) and fed experimental diets described in Table 1 during gestation and lactation periods. We had 2 experimental groups of pregnant rats: CTL group fed with CTL diet and PR group fed with low protein diet. The experimental protocol is schematized in Figure 1.

Pregnant dams were weighted three times a week, and food and water intakes were measured at the same times. At birth, litter size was adjusted to 8 pups per dam (4 females and 4 males). Mortality at birth was calculated on 50–56 pups per sex for 13 litters per group in CTL and PR groups. Mothers and pups were weighed three times a week, and food and water intakes were measured at the same times. At weaning, females and males were separated, housed 4 animals per cage, and fed a breeding diet (A03, SAFE, Augy, France) until postnatal day 50 (PND50). Then, rats were housed 3 per cage and fed a standard diet (A04, SAFE, Augy, France) until the end of the experiments, i.e., PND100. At PND21, mothers were anesthetized with isoflurane 4% and decapitated. Liver, mesenteric white adipose tissue (WAT), perirenal WAT, and mammary gland were weighed. Same tissues were also weighed in pregnant dams at G17 [31]. Tissue samples were snap frozen and kept at -80 C until analysis. For each experimental procedure, a maximum of 2 animals per sex from the same litter was used.

*Insulin and glucose tolerance tests.* Insulin (0.75 U/kg) was injected intraperitoneally in 5 h-fasted animals for insulin tolerance test. Glucose (2 g/kg) was given *per os* in 16 h-fasted animals for oral glucose tolerance test. In both tests, blood glucose was measured at 0, 15, 30, 45, 60, 90 and 120 min, and for OGTT, blood samples were taken at 0, 15, and 30 min. Tests was done at PND30 and PND100 on males and females.

*Lipid preload test.* In female offspring, we assessed the capacity of lipids to alter food behaviour at PND100. To that end, animals were food deprived at 8 a.m. for 5 h. One weighed bottle containing 50 mL of water or Intralipid^®^ 20% (IL; Fresenius Kabi France, Sèvres, France) diluted at 5% with tap water was presented to each animal for 15 min at 1 p.m. Then, a weighted amount of food (A04) was given to animals, and food intake was measured by hand-weighing at 2, 4, and 6 p.m. (i.e., 1 h, 3 h, and 5 h food intake) until the next morning (8 a.m. i.e., 19 h food intake). This protocol was repeated for seven days in a row. The eighth day, water or IL was presented for 1 h without refeeding, followed by transcardiac perfusions performed with saline, then 4% paraformaldehyde on anaesthetized animals. Brains were collected, postfixed in paraformaldehyde, and frozen.

*Biological sampling collection.* Blood samples from pregnant and lactating rats were collected on alert animals by a tail snip in EDTA-tubes at G7, G14, G19, lactation day (L) 6, L13, and L19. Blood samples were centrifuged at 10,000 rpm for 2 min at 4 °C, and plasma was collected in Eppendorf before storage at −20 °C until analysis. At sacrifice, blood samples were collected by an intra-cardiac puncture in EDTA tubes (Pfizer-Centravet, Plancoët, France), before sacrifice on anaesthetized animals, and were treated as described above.

*Triglyceride and cholesterol content in liver.* Around 50 mg of piece of liver were crushed in saline with a zirconium marble bead in PreCellys. A mixture of chloroform:methanol (2:1) was added, then tubes were centrifuged 15 min at 2500× *g*. The organic phase was withdrawn and evaporated under the fume cupboard during few hours. The bottom was resuspended in triton:isopropanol X-100 (9:1) and assayed with colorimetric test for triglyceride and cholesterol detection (Triglycerides FS^®^; Cholesterol FS^®^; Diasys, Holzeim, Germany).

*Biochemical analysis.* Blood glucose was measured on glucometer (ACCU-CHEK^®^ PERFORMA, Roche Diagnostics France, Meylan, France) by a tail snip. Plasma insulin, leptin, and adiponectin were assayed by ELISA (ALPCO, Salem, MA, USA); plasma triglycerides and total cholesterol were assayed by colorimetric tests cited above, following manufacturers’ instructions.

*Immunohistochemistry for c-fos staining.* Thirty-micrometer coronal sections were realized between −0.60 and −4.08 mm of Bregma coordinates, according to Paxino’s atlas [32], with a HM560 cryotome (MM, Francheville, France), collected in cryoconservation solution, and stored at −20 °C until further use. Free-floating sections were rinsed in PBS and exposed to 0.3% hydrogen peroxide for 30 min. They were then preincubated in PBS containing 3% normal goat serum and 0.25% Triton X-100 (blocking solution) for 2 h and incubated overnight at 4 °C under agitation with rabbit polyclonal anti-c-fos antibody (1:5000; sc-52; Santa Cruz Biotechnology, Heidelberg, Germany) in blocking solution. Subsequently, sections were incubated with biotinylated goat anti-rabbit IgG (1:1000; A24541; ThermoFisher Scientific; Waltham, MA USA) in blocking solution for 2 h and with VECTASTAIN^®^ Elite ABC-HRP kit (PK-6100, Vector Laboratories; Burlingame, California, USA) for 30 min. C-fos positive cells were visualised using diaminobenzidine substrate kit with nickel (SK-4100; Vector Laboratories). Several PBS rinses were carried out between the above steps, except between blocking and incubation with primary antibody. Sections were mounted on slides and air-dried. A counterstain was done with methyl green (H-3402; Vector Laboratories) to visualize nuclei, then sections were dehydrated in alcohol and coverslipped with mounting media (VectaMount^TM^; H-5500; Vector Laboratories). Slides were scanned with Hamamatsu (Hamamatsu NanoZoomer 2.0-HT, Hamamatsu Photonics France SARL, Massy, France).

*Counting the c-fos immunoreactive cells.* Number of c-fos-immunoreactive cells were counted bilaterally in different cerebral regions by using a computerized image analysis (Image J). Between 3 and 8 sections per region were analysed. Results were expressed as the mean of the sum of c-fos-positive nuclei counted per µm^2^ in each region of interest. This quantification was made for the arcuate nucleus (ARC), ventromedian nucleus (VMN), paraventricular nucleus (PVN), lateral hypothalamus (LH), and dorsomedian nucleus (DMN).

*Expression of data and statistical analysis.* Data are expressed as individual values with mean ± SEM. Statistical analyses were performed using Student t-test or ANOVA, followed by two-by-two comparisons using Sidak post hoc test (GraphPad Prism 7 version 7.0a, software). Normal distribution and homogeneity of variance were first tested, respectively, with Shapiro–Wilk normality test and F test. *p* < 0.05 was considered significant.

## 3. Results

### 3.1. Maternal PR Does Alter Maternal Food Intake, Body Weight Gain, and Hormonal Status during Gestation and Lactation

Body weight gain during gestation was similar between the CTL and PR groups (Figure 2a).

After delivery, the body weight of CTL mothers remained stable, while PR mother body weight decreased until weaning (CTL: 2.6 ± 3.6 g; PR: −35.1 ± 7.6 g, *p* < 0.001). Food intake was slightly increased during the first days of gestation for PR mothers and was significantly decreased at the end of lactation (Figure 2b). As the low-protein diet contains half as much protein as the control diet, protein consumption clearly decreased in PR mothers (Figure 2c). For PR mothers, lipid and carbohydrate consumption was significantly higher during gestation, with a drop at the end of the lactation period (Figure 2d,e). Interestingly, water intake was decreased in the PR group all along gestation and lactation (Figure 2f).

The profile of blood glucose and hormones normally varies in the course of gestation and lactation. Blood glucose decreased during gestation, remained stable during lactation (Figure 3a), and was on average slightly higher in PR mothers (CTL: 5.8 ± 0.1 mM; PR: 6.7 ± 0.3 mM, *p* < 0.05). Non-fasting plasma insulin levels increased during gestation and dramatically dropped during lactation for both the CTL and PR groups (Figure 3b). Plasma leptin was higher in PR mothers at the end of gestation (*p* < 0.001; Figure 3c) and remained higher during all time points of the lactation period. Plasma adiponectin levels increased significantly during gestation and remarkably during lactation in PR mothers, reaching three times more plasma adiponectin in PR mothers compared to CTL mothers in late lactation (Figure 3d). Plasma triglycerides increased significantly in PR mothers before birth and were similar to the CTL group during lactation (Figure 3e). Plasma total cholesterol and non-esterified fatty acids were quantified during gestation, with a slight increase of plasma total cholesterol in PR mothers in the mid-gestation and a slight increase of non-esterified fatty acids in late gestation.

Liver weight increased in the same manner in both the CTL and PR groups from G17 to L21 (Figure 3f). Mammary gland weight per body weight was not significantly affected by the maternal diet (Figure 3g). Perirenal WAT per body weight tended to be higher in PR mothers at the end of gestation (Figure 3h), while mesenteric WAT was similar between the groups (Figure 3i). Triglyceride and cholesterol hepatic contents were increased in PR livers (*p* < 0.05 for triglycerides, Figure 3j).

### 3.2. Despite Unchanged Birth Weight, Maternal PR Has Long Lasting Effects after Birth on the Body Weight of the Offspring

At birth, the number of pups by litter and the sex ratio were similar between the CTL and PR groups (Table 2). The weight was similar when we compared the litter or male and female body weight. Maternal diet did not influence birth mortality.

Male growth was severely reduced during the lactation period (Figure 4a), resulting in decreased body weight gain at weaning (CTL: 53.6 ± 0.9 g; PR: 31.2 ± 0.7 g; *p* < 0.001; Figure 4b) that persisted at least until PND100 (CTL: 563.6 ± 27.1 g; PR: 490.3 ± 12.7 g; *p* < 0.05; Figure 4c,d). We observed the same evolution in female offspring, with a lower body weight gain at weaning (CTL: 53.3 ± 0.9 g; PR: 31.0 ± 0.6 g; *p* < 0.001; Figure 4e,f) and at PND100 (CTL: 315.3 ± 6.8 g; PR: 290.3 ± 3.5 g; *p* < 0.01; Figure 4g,h). The perirenal WAT was also decreased in male and female offspring PR at PND21 and PND100.

### 3.3. Maternal PR Has a Small but Significant Impact on Glucose Tolerance in Young (PND30) Male and Female Offspring without Significantly Affecting Insulin Secretion

At PND30, insulin sensitivity tested by the insulin tolerance test did not differ between the groups (Figure 5a,b), although the basal blood glucose was significantly increased in male offspring (*p* < 0.05; unpaired *t*-test; Figure 5a). Glucose tolerance was not modified in PR males, despite a reduced area under the curve (AUC) and a slightly lower insulin secretion in response to glucose (Figure 5c–e).

In females, basal blood glucose was similar between the groups, and glucose tolerance was better in PR females (Figure 5f), with an AUC value significantly lower (Figure 5g) despite no change in insulin secretion (Figure 5h).

At PND100, glucose tolerance was similar in PR males (Figure 5i,j) and females (Figure 5l,m), despite significantly lower plasma insulin in both sexes (Figure 5k,n).

### 3.4. Maternal PR Modifies Lipid Sensing in Female Offspring at PND100

During the training period (Figure 6a), female offspring were trained to consume a load of IL whose intake reached a plateau around 50 kcal/kg/day at D4 and later (Figure 6b), suggesting that there was no difference in fat liking between groups.

Once rats were habituated, we measured chow pellets consumption after a preload of water or IL during each day from D4 to D7 over a period of 1 h, 5 h, and 19 h in order to distinguish the early or late response to the preload. Water preload did not modify the first hour of food intake between groups (Figure 6c), whereas IL preload increased food intake during the first hour for the CTL group, but not for the PR females (*p* < 0.05; 2-way ANOVA; Figure 6d). The overall consumption from D4 to D7 did not show significant differences (Figure 6e). Over the 5 h period, CTL rats presented a significant increase of food intake between D1 and D4 (*p* < 0.05; 2-way ANOVA followed by Tukey’s test), while the food intake of PR females was not changed when the preload was water (Figure 6f). After the IL preload, the 5 h period of food intake was significantly increased over time (*p* < 0.05; 2-way ANOVA) but not different between the CTL and PR groups (Figure 6g). AUC tended to be lower for PR females after water preload and was not changed after IL preload (Figure 6 h). Over the 19 h period, food intake was significantly increased after water or IL preload in both groups (*p* < 0.05; 2-way ANOVA; Figure 6i,j). AUC was higher for PR females after water preload and tended to be lower after IL preload (*p* < 0.05; 1-way ANOVA; Figure 6k). This suggests that the first phase of lipid sensing seemed more efficient in PR females. In addition, after lipid preload, the third hour of food intake was more important in PR females (*water*: 2.7 ± 1.1%; *IL*: 23.3 ± 4.6% of the total food consumption; *p* < 0.05; Figure 6m) than in CTL females (*water:* 5.1 ± 2.6%; *IL*: 13.2 ± 5.5% of the total food consumption; non-significant; Figure 6l), suggesting that the satiety period was reduced in the PR group.

### 3.5. Maternal PR Changes the Number of c-fos Immunoreactive Cells in Hypothalamus in Basal State

As we observed that food behaviour is affected in PR females, we performed c-fos staining in several hypothalamic areas involved in food intake control after a 5 h fast, i.e., basal state (Figure 7a–e), and 1 h exposure to water or IL to assess the effect of lipids on the activity of hypothalamic cells (Figure 7f–k).

At basal state, the number of c-fos-positive cells differed only in the VMN, with a higher number in PR females (*p* < 0.05; Figure 7e). The 1 h IL consumption, similar between CTL and PR females (CTL: 75.6 ± 15.2 kcal/kg; PR: 71.0 ± 5.8 kcal/kg), induced a c-fos-positive signal in the LH area of PR females that tended to be higher than in the basal state (*p* = 0.08; Figure 7i).

## 4. Discussion

This present study focused on the effect of maternal PR during gestation and lactation on maternal parameters and offspring growth, metabolism, and food behaviour. There is considerable evidence that PR alters the hormone profile and lipid and glucose metabolism, with some discrepancies in the consequences for offspring feeding behaviour. In addition, few studies have focused on the health of the mother during gestation and lactation and on the consequence on female offspring, as most studies were performed on male offspring only.

Increased maternal food intake after PR during gestation has been well documented and has emerged as a mechanism to compensate for the lack of amino acid supply [33,34,35,36]. We showed that food intake was drastically decreased during the second half of lactation, which could deeply alter breast milk quality and body composition, given the high energy requirements for nursing [37]. In addition, we showed that water intake was reduced in PR mothers during gestation and lactation, and this is probably the reason why milk production is reduced in maternal restriction models, as observed in our laboratory [36,38]. Indeed, the higher the protein content in the diet, the higher the water consumption because protein intake is a modulator of renal function [36,39,40].

Low protein intake below the levels required to maintain the nitrogen balance has major effects on the control of food intake and on body composition and induces accumulation of body fat in adult rats [35]. In our model, we detected more perirenal WAT in PR mothers at the end of gestation and higher triglyceride hepatic content. As we did not observe any body weight gain differences between the two groups during gestation, it is possible that body fat mass increased at the expense of lean mass in PR mothers, which is in agreement with previous observations [34,35]. In addition, PR mothers lost more weight after delivery, without fat loss of perirenal and mesenteric fat pads. The very high energy and amino acids requirements for pup growth may have induced protein catabolism and lean-mass loss in PR mothers. This could be investigated by amino acids turnover in maternal and fetal muscle.

The greater fat mass could explain the greater plasma leptin and adiponectin levels observed in PR mothers during late gestation and lactation periods, as well as plasma and hepatic triglyceride levels. A similar observation was made on PR rats where serum leptin and adiponectin were increased, as well, along with hepatic triglyceride content, and were positively correlated with body fat content [34,41]. Overall, these findings may reflect an adaptative mechanism of leptin and adiponectin resistance to facilitate energy intake and compensate for low protein intake. In the context of pregnancy, central and placental leptin resistance occurs to counteract the satietogenic effect of leptin [42], and limited evidence also suggests some degree of adiponectin resistance [43]. Thus far, few studies investigated maternal leptin and adiponectin levels in the context of PR. In animal models, altered leptin signalling during gestation may predispose the fetus to leptin resistance and obesity development in later life [44,45]. In humans, pregnancies complicated by IUGR are associated with an elevated maternal blood leptin concentration at delivery, which is concomitant with a lower concentration of leptin in cord blood [7]. Information regarding maternal adiponectin in the context of IUGR is lacking. However, it is known that obesity is associated with a reduced plasma adiponectin in obese animals, humans, and patients with type 2 diabetes. Improving maternal adiponectin levels in obese mice mothers may be an effective intervention strategy to prevent fetal overgrowth and intrauterine transmission of obesity and metabolic disease to the next generation [46].

Early changes in leptin and adiponectin concentrations have been shown in LBW infants, partially influenced by the maternal pre-gestational body mass index and offspring postnatal weight gain [6]. In our model, we did not measure plasma leptin and adiponectin levels in the offspring, making it impossible to predict whether the elevated maternal leptin and adiponectin levels were transmitted to the offspring through the placenta or milk, despite the fact that variations in the hormonal contents of maternal plasma are known to influence maternal milk hormones in rodents and humans [8,22,47].

Earlier studies performed in our lab and by others demonstrated variable changes in birth weight, body weight gain, and energy homeostasis in animal models of perinatal maternal caloric restriction or PR [16,48,49,50,51]. Contradictory findings between studies could be due to differences in diet composition, especially the nature and quantity of carbohydrates, i.e., corn starch or sucrose, that often vary according to the percentage of proteins. Glucose intolerance has been reported in young male rats (PND30), accompanied by a defect in glucose-induced insulin secretion in a model of prenatal maternal caloric restriction [49]. Our present experiment did not reveal major modifications in glucose tolerance in male and female offspring under a chow diet, except for a slight increase at PND30 in females. At PND100, insulin secretion in response to glucose was lower in both males and females, which could reflect the early onset of insulin secretion failure observed later in life in PR male offspring at PND234 and exacerbated by a high-energy diet [48].

Alterations of eating behaviour have been shown in models of IUGR [23,52,53] and in humans [10,11,12], but little information is generally available on female offspring. In our study, CTL and PR females had similar appetite for lipids when presented as triglyceride emulsion (i.e., Intralipid^®^). We previously observed differences in the food preferences of male rats from PR mothers when faced with a choice between standard, fatty, and sweet foods [22]. In addition, it has also been demonstrated that, in the case of maternal food restriction during gestation, male offspring have a greater appetite for palatable food, which can be attributed to a delay of the dopaminergic response to palatable foods [53]. In humans, IUGR can cause persistent and sex-dependant changes in the preference for palatable foods [10,11,12,13]. In our model, we tested the lipid sensing and its consequences on food intake and satiety. We have previously shown that a short-term lipid infusion toward the brain decreases food intake in Wistar rats [26], and it was also known that intestinal fat infusions decrease food intake in pigs and humans [54,55,56]. In our protocol, the IL given *per os* was far less efficient for inhibiting food intake. However, we were able to demonstrate that the early phase of lipid sensing (after 1 h) was more efficient in the PR group for regulating food intake, but that a decay in the satiety was visible at 3 h. It is possible that under this route of administration, intestinal lipid sensing was more involved in the response to lipids than hypothalamic sensing, as evidenced by the low c-fos cells activation we measured. The IL c-fos-induced signal in the brain tended to be higher in the LH nucleus. LH is a crucial hypothalamic area that detects available nutrients and also receives neuronal sensory inputs and reward/motivation-related information from other brain areas. Through its integration capacity and its projections to the brain stem and the spinal cord, it actively participates in the regulation of energy homeostasis [14].

## 5. Conclusions

The present study provides new data on the consequences of maternal PR during gestation and lactation on maternal leptin and adiponectin hormone levels that may explain some of the alterations in metabolism and food behaviour demonstrated in the offspring. In addition, some differences in the effect of an IL preload on food intake and satiety response between PR and CTL females reinforce the idea of an early programming of feeding behaviour induced by low maternal protein intake during gestation and lactation, although the mechanisms need further investigation.

## Figures and Tables

**Figure 1 nutrients-15-00463-f001:**
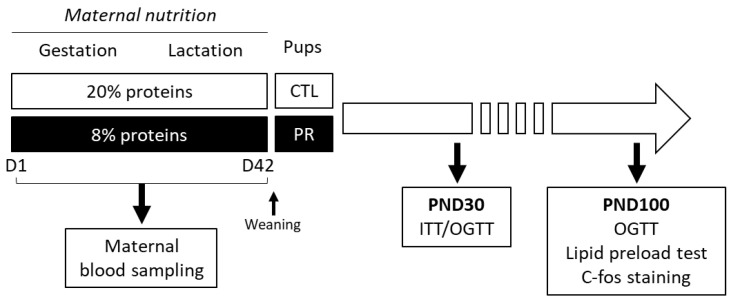
Schematic diagram of the study design. Two groups of animals were generated from mothers fed with control diet (CTL; 20% proteins) or low-protein diet (PR; 8% proteins). Insulin sensitivity and glucose tolerance were studied in offspring at postnatal day (PND) 30 and PND100. Lipid preload test and c-fos staining in brain were performed in female offspring at PND100 to test the impact of maternal diet on hypothalamic lipid sensing in offspring. ITT, insulin tolerance test; OGTT, oral glucose tolerance test.

**Figure 2 nutrients-15-00463-f002:**
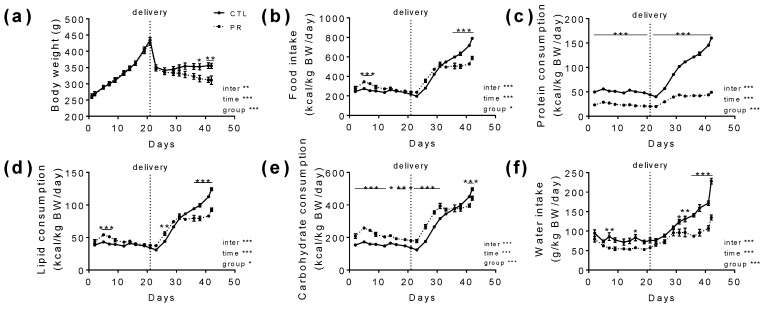
Dams follow-up during gestation and lactation. Body weight (**a**); food intake (**b**); protein (**c**), lipid (**d**), and carbohydrate (**e**) consumption; and water intake (**f**) are presented during gestation (from D1 to D21) and lactation (from D22 to D42). Solid line, CTL; Dotted line, PR. Vertical dotted line at D21 indicates delivery. CTL, control. PR, protein restriction, BW, body weight. Data are mean ± SEM (*n* = 8–13). * *p* < 0.05, ** *p* < 0.01, *** *p* < 0.001 vs. CTL, 2-way ANOVA with Sidak post-test.

**Figure 3 nutrients-15-00463-f003:**
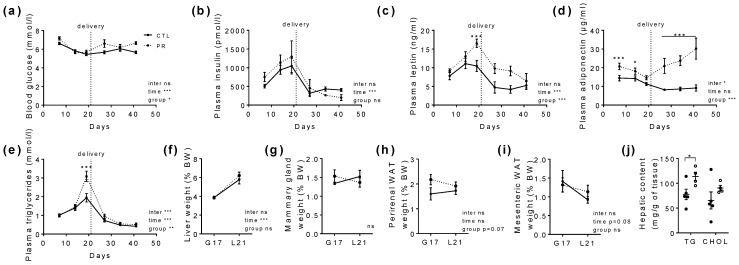
Plasma parameters and adiposity follow-up of dams during gestation and lactation. Blood glucose (**a**), plasma insulin (**b**), leptin (**c**), adiponectin (**d**), and triglyceride (**e**) levels have been followed during gestation (from D1 to D21) and lactation (from D22 to D42). At weaning, mothers have been sacrificed and liver (**f**), mammary gland (**g**), perirenal (**h**), and mesenteric (**i**) white adipose tissues have been weighed. Triglyceride and cholesterol contents (**j**) have been assayed in the liver. Solid line, CTL; Dotted line, PR. Vertical dotted line at D21 indicates delivery. CTL, control. PR, protein restriction; WAT, white adipose tissue; TG, triglycerides; CHOL, cholesterol. Data are mean ± SEM or individual values (*n* = 2–13). * *p* < 0.05, ** *p* < 0.01, *** *p* < 0.001 vs. CTL, 2-way ANOVA with Sidak post-test.

**Figure 4 nutrients-15-00463-f004:**
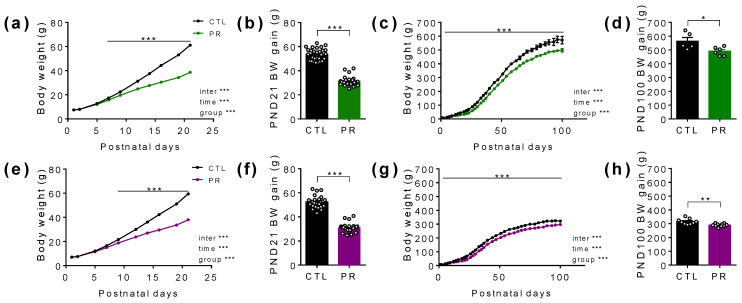
Postnatal growth of male and female offspring. Body weight (BW; **a**,**c**,**e**,**g**) and BW gain at postnatal day (PND) 21 (**b**,**f**) and PND100 (**d**,**h**) have been followed in male (**a**–**d**) and female (**e**–**h**) offspring from the birth. Black line, CTL; coloured line, PR. Green, male; purple, female. CTL, control. PR, protein restriction. Data are mean ± SEM or individual values (*n* = 5–31). * *p* < 0.05, ** *p* < 0.01, *** *p* < 0.001 vs. CTL, 2-way ANOVA with Sidak post-test, unpaired *t*-test.

**Figure 5 nutrients-15-00463-f005:**
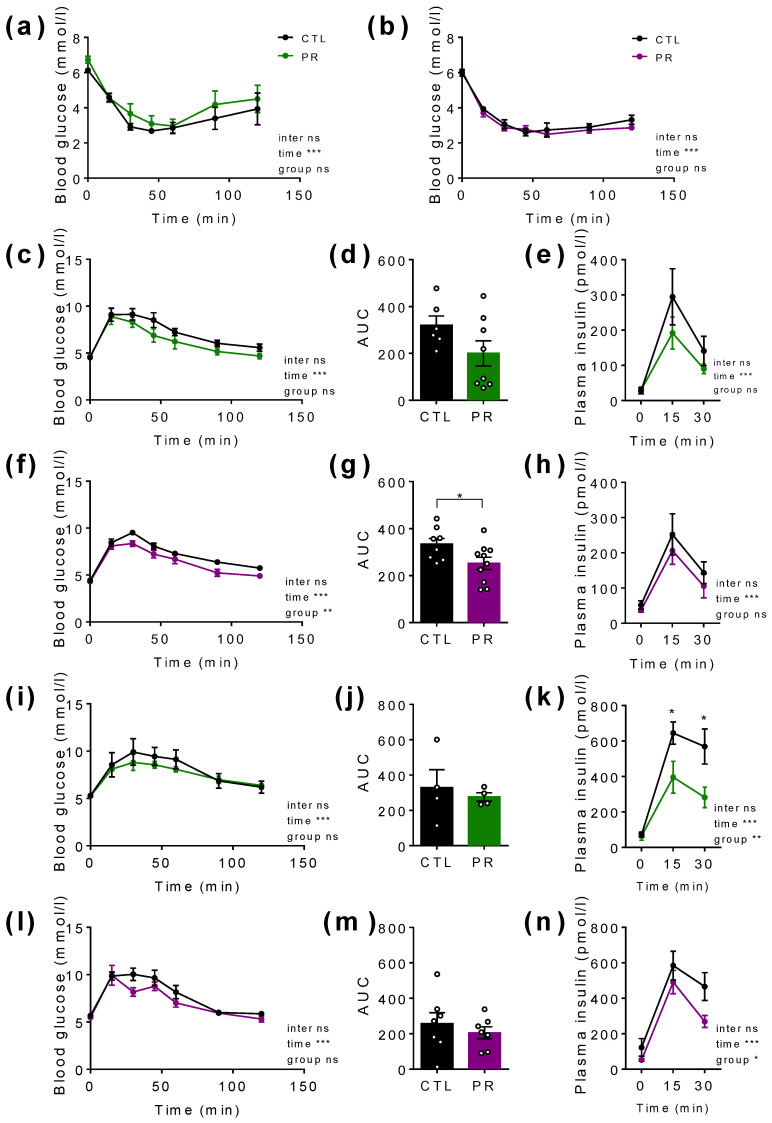
Insulin and glucose tolerance in male and female offspring at postnatal day (PND) 30 and PND100. Blood glucose follow-up after intraperitoneal insulin injection (0.75 U/kg) in male (**a**) and female (**b**) after 5 h-fasting. Blood glucose follow-up after oral glucose gavage (2 g/kg) in male (**c**,**i**) and female (**f**,**l**) at PND 30 (**c**–**h**) and PND100 (**i**–**n**). Area under the curve (AUC) for male (**d**,**j**) and female (**g**,**m**). Plasma insulin in response to blood glucose increase in male (e,k) and female (**h**,**n**). Black line, CTL; coloured line, PR. Green, male; purple, female. CTL, control. PR, protein restriction; AUC, area under the curve. Data are mean ± SEM or individual values (*n* = 5–10). * *p* < 0.05, ** *p* < 0.01, *** *p* < 0.001 vs. CTL, 2-way ANOVA with Sidak post-test, unpaired *t*-test.

**Figure 6 nutrients-15-00463-f006:**
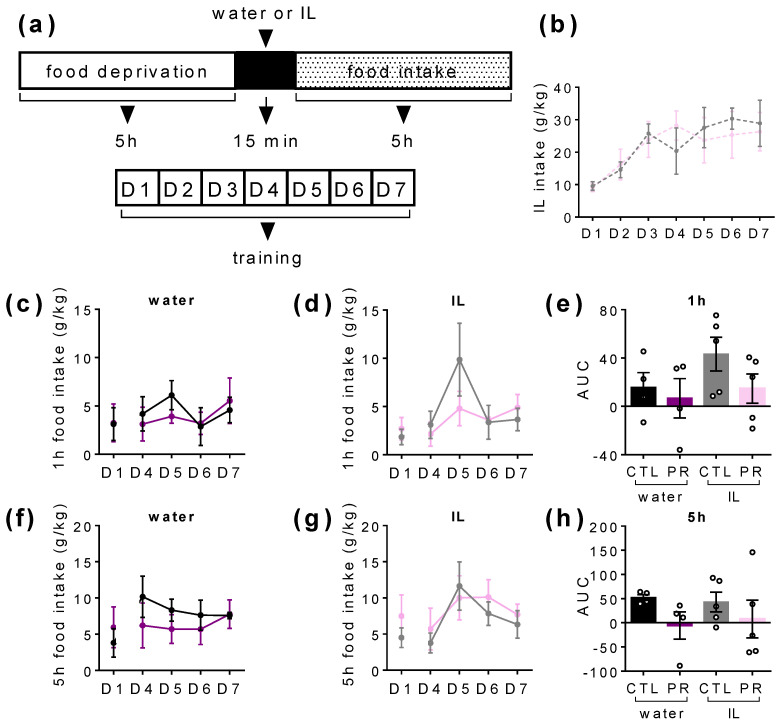
Food intake measurement after water or Intralipid 5% (IL) ingestion in female offspring at postnatal day (PND) 100. Schematic representation of the experiment (**a**): after 5 h-food deprivation, a weighed bottle of water or IL was presented to rats during 15 min then removed. Food pellets were given back and food intake was measured after 1 h, 3 h, 5 h, and 19 h. This sequence was repeated 7 days in a row. IL intake (**b**) each day of the training period. Daily 1 h-food intake after water (**c**) and IL (**d**) preload, and AUC (**e**) in CTL and PR females. Daily 5 h-food intake after water (**f**) and IL (**g**) preload, and AUC (**h**) in CTL and PR females. Daily 19 h-food intake after water (**i**) and IL (**j**) preload, and AUC (**k**) in CTL and PR females. The 1 h-, 3 h-, and 5 h-calorie intakes in CTL and PR females after water (**l**) or IL preload (**m**). Black and purple bars, water; grey and pink bars, IL. Solid line, food intake. Dotted line, IL intake. CTL, control. PR, protein restriction. IL, intralipid. AUC, area under the curve. Data are individual values and mean ± SEM or individual values (*n* = 4–5). * *p* < 0.05 vs. CTL, 1-way or 2-way ANOVA with Sidak post-test, unpaired *t*-test.

**Figure 7 nutrients-15-00463-f007:**
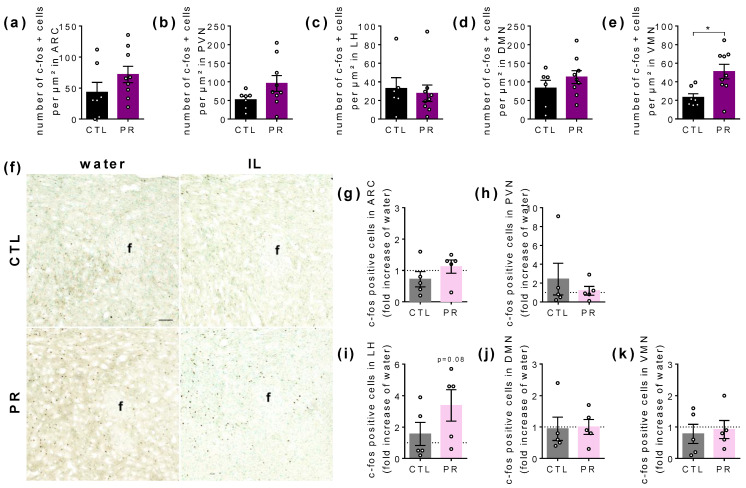
Immunostaining of c-fos-positive cells in hypothalamic nuclei after 1 h water or Intralipid 5% (IL) ingestion in female offspring at postnatal day (PND). Number of c-fos-positive cells in ARC (**a**), PVN (**b**), LH (**c**), DMN (**d**), and VMN (**e**) in CTL and PR females. Representative photomicrographs of lateral hypothalamus after water or IL ingestion in CTL and PR females (**f**). C-fos-positive cells in ARC (**g**), PVN (**h**), LH (**i**), DMN (**j**), and VMN (**k**) in fold increase of water CTL and PR females. Black and purple bars, water; grey and pink bars, IL. CTL, control. PR, protein restriction. IL, intralipid; ARC, arcuate nucleus; PVN, paraventricular nucleus; LH, lateral hypothalamus; DMN, dorsomedian nucleus; VMN, ventromedian nucleus; f, fornix. Scale bar = 100 nm. Data are individual values and mean ± SEM or individual values (*n* = 4–5). * *p* < 0.05 vs. CTL, unpaired *t*-test.

**Table 1 nutrients-15-00463-t001:** Composition of control and low-protein diets.

g/100 g	Control Diet	Low Protein Diet
Casein	20	8
L-Cystine	0.3	0.3
Corn starch	39.75	47.32
Maltodextrine	13.20	15.72
Sucrose	10	11.91
Cellulose	5	5
Corn oil	7	7
t-butylhydroquinone	0.0014	0.0014
Mineral mix AIN 93 G ^a^	3.5	3.5
Vitamin mix AIN 93 Vx ^b^	1	1
Choline bitartrate	0.25	0.25
Total calories	3948	3959

^a^ Mineral mix AIN 93 G = Calcium Carbonate 35.7%, Monopotassium Phosphate 19.6%, Potassium Citrate Monohydrate 7.078%, Sodium Chloride 7.4%, Potassium Sulfate 4.66%, Magnesium Oxide 2.4%, Ferric Citrate 0.606%, Zinc Carbonate 0.165%, Manganese Carbonate 0.063%, Copper Carbonate 0.03%, Potassium Iodate 0.001%, Sodium Selenate, Anhydrous 0.00103%, Ammonium Molybdate.4H_2_O 0.000795%, Sodium Metasilicate.9H_2_O 0.145%, Chromium Potassium Sulfate.12H_2_O 0.0275%, Lithium Chloride 0.00174%, Boric Acid 0.008145%, Sodium Fluoride 0.00635%, Nickel Carbonate 0.00318%, Ammonium Vanadate 0.00066%, Powdered Sugar 22.1%. ^b^ Vitamin mix AIN 93 Vx (gm/kg) = Nicotinic Acid 3.00, D-Calcium Pantothenate 1.60, Pyridoxine HCl 0.70, Thiamine HCl 0.60, Riboflavin 0.60, Folic Acid 0.20, D-Biotin 0.02, Vitamin B12 (0.1% triturated in mannitol) 2.50, a-Tocopherol Powder (250 U/gm) 30.00, Vitamin A Palmitate (250,000 U/gm) 1.60, Vitamin D3 (400,000 U/gm) 0.25, Phylloquinone 0.075, Powdered Sucrose 959.655.

**Table 2 nutrients-15-00463-t002:** Parturition.

At Birth		CTL	PR
Pup number	Total	11.5 ± 0.3	11.3 ± 0.5
	Male	5.7 ± 0.5	6.4 ± 0.4
	Female	5.8 ± 0.6	4.9 ± 0.5
Mortality at birth (%)		1.2 ± 1.2	0.0 ± 0.0
Litter weight (g)		79.4 ± 2.3	81.9 ± 3.4
Pup weight (g)	Male	7.5 ± 0.1	7.4 ± 0.1
	Female	7.1 ± 0.1	7.0 ± 0.1

*n* = 50–56 pups per sex for 13 litters per group. Values are mean ± sem. CTL, control diet. PR, protein restriction diet. Student *t*-test.

## Data Availability

Not applicable.

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
