# Peer review of "Maternal Protein Restriction in Rats Alters Postnatal Growth and Brain Lipid Sensing in Female Offspring"

_nutrients, 2023, doi:10.3390/nu15020463_

Round 1

Reviewer 1 Report

This study was designed to investigate the impact of maternal protein restriction during gestation and lactation on glucose homeostasis and eating behavior in female offspring. It was concluded that maternal protein restriction during gestation and lactation does induce extrauterine growth restriction, accompanied by alterations in maternal plasma leptin and adiponectin levels which may be involved in programming the alterations in eating behavior observed in females at adulthood. This is a very interesting discovery, however, the study was not well designed, paper was not well written, and the data was not well presented.

The followings are a few example of the pitfalls.

1. The abbreviations, like "CTL" and "PND10" (Line 16, Page 1), should be spelled in full in the first mention.

2. The total amount of each component in the Control diet (g/100g) contains 102.5214 g, while Low protein diet (g/100g) contains 100.0014 g in the Table 1.

3. In the statistical analysis section, analysis of normality and homogeneity of variance should be performed before ANOVA analysis. Please consult professional statistician to revise this section.

4. The font size in many figures are too small to read.

5. The discussion section is too long, which should be shortened and focus on the main findings of this study.

Reviewer 2 Report

1.       The paper is well written, but difficult to read, because of the rather complex experimental protocol.

Therefore, it would be a benefit for the reader to display an infographic in the abstract or in the introduction, which shows the different steps of the experimental protocol and procedure as well as determinations.

2.       In the legend Figure 3, BW and PND need to be explained

3.       Same in the legend of figure 4 and 5

4.       It would be nice to have histological picture of one of the immunostainings of figure 5, a and f to show visually the difference between the feeding regimens of CTL and RP in the brain, because this is one of the main findings.
